# Severe COVID-19 Complicated by Cerebral Venous Thrombosis in a Newborn Successfully Treated with Remdesivir, Glucocorticoids, and Hyperimmune Plasma

**DOI:** 10.3390/ijerph182413201

**Published:** 2021-12-15

**Authors:** Laura Cursi, Francesca Ippolita Calo Carducci, Sara Chiurchiu, Lorenza Romani, Francesca Stoppa, Giulia Lucignani, Cristina Russo, Daniela Longo, Carlo Federico Perno, Corrado Cecchetti, Mary Haywood Lombardi, Patrizia D’Argenio, Laura Lancella, Stefania Bernardi, Paolo Rossi

**Affiliations:** 1Academic Department of Pediatrics, Bambino Gesù Children’s Hospital—IRCCS, 00165 Rome, Italy; sara.chiurchiu@opbg.net (S.C.); Lorenza.romani@opbg.net (L.R.); patrizia.dargenio@opbg.net (P.D.); laura.lancella@opbg.net (L.L.); Stefania.bernardi@opbg.net (S.B.); paolo.rossi@opbg.net (P.R.); 2Department of Emergency, Acceptance and General Pediatrics, Bambino Gesù Children’s Hospital, IRCCS, 00165 Rome, Italy; Francesca.stoppa@opbg.net (F.S.); corrado.cecchetti@opbg.net (C.C.); maryhaywood.lombardi@opbg.net (M.H.L.); 3Imaging Department, Bambino Gesù Children’s Hospital, IRCCS, 00165 Rome, Italy; giulia.lucignani@opbg.net (G.L.); daniela.longo@opbg.net (D.L.); 4Department of Laboratories, Bambino Gesù Children’s Hospital, IRCCS, 00165 Rome, Italy; cristina.russo@opbg.net (C.R.); carlofederico.perno@opbg.net (C.F.P.)

**Keywords:** newborn, SARS-CoV-2, COVID-19, cerebral thrombosis, remdesivir

## Abstract

Severe acute respiratory syndrome coronavirus-2 (SARS-CoV-2) is responsible for the coronavirus disease 2019 (COVID-19) pandemic, affecting all age groups with a wide spectrum of clinical presentation ranging from asymptomatic to severe interstitial pneumonia, hyperinflammation, and death. Children and infants generally show a mild course of the disease, although infants have been observed to have a higher risk of hospitalization and severe outcomes. Here, we report the case of a preterm infant with a severe form of SARS-CoV-2 infection complicated by cerebral venous thrombosis successfully treated with steroids, hyperimmune plasma, and remdesivir.

## 1. Introduction

Severe acute respiratory syndrome coronavirus-2 (SARS-CoV-2) is responsible for the coronavirus disease 2019 (COVID-19) pandemic, affecting all age groups with a wide spectrum of clinical presentation ranging from asymptomatic to severe interstitial pneumonia, hyperinflammation, and death [1]. Children and infants generally show a mild course of the disease, although infants have been observed to have a higher risk of hospitalization and severe outcomes [2]. Neonatal SARS-CoV-2 infections are rare, despite the fact that newborns are at risk for vertical and postpartum horizontal transmission [3,4], but they can have a severe clinical course, especially in preterm infants. Here, we report the case of a preterm infant with a severe form of SARS-CoV-2 infection complicated with cerebral venous thrombosis.

## 2. Case Description

A nine day old male newborn was admitted to our hospital due to fever and poor general condition. The pregnancy was complicated by threatened miscarriage and placental abruption. He was born at 36 weeks + 1 day of GA by spontaneous delivery. Perinatal cardiotocographic monitoring was negative. Neonate blood gas analyses and cardiorespiratory adaption were normal, and the Apgar score was 7 and 8 at 1′ and 5′ minutes, respectively. Birth weight was 2950 g. The subsequent early postnatal period was complicated by transient hypoglycemia; neonatal clinical assessment was normal, postnatal weight loss was within normality range (<10%), and the neonate was discharged on the fourth day of life. The mother was tested for SARS-CoV-2 at admission in the obstetric ward with a negative result and a positive result at discharge, without any symptoms. At day nine, the baby developed fever (38 °C) and poor feeding.

The nasopharyngeal swab, tested for SARS-CoV-2 by qualitative realtime PCR (Allplex^TM^ SARS-CoV-2 Assay, Seegene), was positive; thus, he was admitted to our COVID-19 center. In the subsequent 24 h, he developed progressive respiratory failure and diarrhea with enterorrhagia and was admitted to the PICU. Surgical evaluation with abdominal X-ray and ultrasound excluded the suspicion of volvulus or necrotizing enterocolitis; echocardiography and electrocardiogram were normal although the N-terminal prohormone of brain natriuretic peptide (NT-proBNP) and Troponin T (TnT) were elevated (Table 1).

The baby was supported with noninvasive ventilation, and treatment with antibiotic wide coverage (ampicillin plus gentamycin plus metronidazole) was started. In the subsequent days, the neurological condition deteriorated with impaired consciousness and pathological spontaneous motricity, and we observed worsening of respiratory failure leading to intubation and mechanical ventilation on day three. The chest radiograph and CT scan showed a picture of bilateral interstitial pneumonia with an extensive area of atelectasis in the basal site.

On the nasopharyngeal swabs (NPS) and bronchoalveolar lavage (BAL), SARS-CoV-2 viral load was assessed by quantitative realtime PCR (Quanty COVID-19 assay, Clonit Srl, Milan, Italy), revealing a high viral load at admission (Table 2).

SARS-CoV-2 variant determination was performed by realtime PCR single-nucleotide-polymorphism (SNP) detection approach (COVID-19 Variant Catcher, Clonit Srl, Milan, Italy, CE IVD). N501Y mutation and HV69-70 deletion, suggesting VOC 202012/01 Lineage B.1.1.7 variant (also known as UK variant), were detected. Thus, we started anti SARS-CoV-2 hyperimmune plasma day three and four, plus dexamethasone 0.15 mg/kg/die and remdesivir 2.5 mg/kg day 5 and 1.25 mg/kg for nine more days.

Because of the neurological impairment, a lumbar puncture was performed and was normal, with negative microbiological results including SARS-CoV-2. The metabolic test on blood and liquor were normal. Immunological screening showed no abnormalities. The EEG showed hypovolted brain electrical activity with a discontinuous pattern; no electrographic or electroclinical seizure was recorded. The cerebral CT scan was normal; the brain MRI showed deep medullary vein thrombosis associated with cytotoxic edema in the deep periventricular white matter (fan-shaped configuration) (Figure 1, Figure 2 and Figure 3), thus, treatment with enoxaparin 100 UI/kg q 12 was started.

The clinical condition progressively improved; after thirteen days the baby was extubated and in four more days became oxygen independent, able to feed on the breast, with neurological and EEG improvement. Extended thrombophilic screening including homocysteine, anticardiolipin antibodies, antiphospholipid antibodies, anti-beta-2-glycoprotein antibodies, protein S, protein C, activated protein C resistance, lupus anticoagulant, factor XIII, von Willebrand factor antigen, Factor V Leyden (G1691A) mutation, factor II prothrombin (G20210A), MTHFR (C677T) mutation all were normal. Genetic investigation of CFTR (Cystic Fibrosis Transmembrane conductance Regulator) showed normal results. After 39 days, the nasopharyngeal swab for SARS-CoV became negative, and the baby was discharged after 42 days. At the five month follow-up visit, the baby was doing well, without any clinical problems. A T1-weighted MRI showed significant reduction in linear hyperintense lesions, normal intensity of periventricular white matter, and enlargement of lateral ventricles (Figure 4, Figure 5 and Figure 6).

## 3. Discussion

The present case report describes a complicated form of SARS-CoV-2 infection in a 36 + 1 week old infant characterized by high persistent fever, diarrhea with enterorrhagy, acute respiratory failure secondary to pneumonia, and cerebral venous thrombosis. These clinical conditions according to the WHO classification fall into the severe form [5]. The infection was acquired post-natally through environmental exposure to the mother positive to nasopharyngeal swab in the fourth day post delivery. The literature reports that neonatal SARS-CoV-2 becomes clinically evident in half of the patients as they developed features of COVID-19 [6,7]. The clinical appearance of neonatal COVID-19 seems similar to those reported in older patients both in terms of symptoms and laboratory or imaging abnormalities, and the outcome is generally favorable. Despite these data there is little information on neonatal forms with a severe clinical course and related treatment guidelines [8,9].

In our case, the patient developed an acute early onset of the disease requiring hospitalization and transfer to the PICU in just 24 h. The viral load of the pharyngeal nose swab performed at nine days old (first day of hospitalization) was very high with a value equal to 1.17 × 10^10^ cp/mL (Table 2), and the genotyping highlighted the UK variant of the virus. The UK variant is associated with a higher viral load and greater transmissibility but not with a more serious or fatal course of infection. Patients affected by the UK variant tend to be younger (<60 years) than those of other variants [10]. Brookman et al. found no evidence of more severe disease in children and young people during the second wave, suggesting that infection with the UK variant does not result in an appreciably different clinical course to the original strain. They also found that Severe Acute Respiratory COVID-19 remains an uncommon occurrence in children and young people [11]. In infants, these data are unknown.

The worsening of the clinical conditions in our patient was so rapid that within 48 h of hospitalization enterorrhagy, interstitial pneumonia, impaired consciousness, and pathological spontaneous motricity were manifested. Interstitial pneumonia also led to acute respiratory insufficiency, which required intubation and mechanical ventilation of the patient. The rapid progression of the disease associated with multiorgan failure led us to consider Multisystem Inflammatory Syndrome in Children and adolescents temporally related to COVID-19 (MIS-C) as a differential diagnosis; however, our newborn did not show the significant increase in inflammation markers, which is an essential condition for the diagnosis [12]. Moreover, the high viral load found not only on the nasopharyngeal swab but also on broncoalveolar lavage led clinicians to suggest remdesivir rather than to infuse high dose immunoglobulins as suggested by the guidelines of the American College of Rheumatology for the treatment of MIS-C [13].

Remdesivir (RDV) use is recommended in children above 12 years of age with COVID-19 pneumonia, and it is the only antiviral drug that has shown some effectiveness in clinical trials [14]. Preliminary data from China reported that RDV was not associated with clinical benefits in adult patients treated with the drug [14]. However, recent results of clinical trials including more than 1000 adults observed a shorter time to recovery with RDV compared to a placebo [14]. RDV has received emergency approval for treating COVID-19 in people above 12 years of age, although few data in children are available, as most clinical trials have focused on adult patients [14,15]. In addition, pediatric pharmacokinetics of RDV that analyze the association between drug dose plasma exposure and intracellular drug exposure are currently unavailable [16].

On the third day, we asked for compassionate use of RDV; the overall time to accomplish the legal requirement for compassionate use (approval from the company, from the local ethics committee, and from the parents) took two days, and the infusion of the drug began on the fifth day of illness at the dosage of 2.5 mg/Kg/day in the first day and 1.25 mg/Kg/day from the second day.

Due to the serious general condition of the patient, on the third day of hospitalization we requested the Ethics Committee for the use of COVID-19 convalescent plasma (CP).

Following the positive response along with parental consent, on the fourth day a dose of 10 mL/kg for two consecutive days was administered. The goal was to support the immature immune system of the newborn by reducing the replication of the virus and thereby improving the patient’s clinical condition.

CP has been cited in several works and proposed as one of the treatments for COVID-19, especially for severe and critical cases with rapid disease progression [17,18,19]. Although insufficient data are available to demonstrate mortality reduction, some N-RCTs and case reports have found that the CP can help to improve clinical symptoms and clear the virus, in particular, if administered within 10 days of onset of disease [17].

The treatment was well tolerated without the appearance of side effects, but at the end of the infusion, there were no substantial changes both clinical and viral load on BAL. For this reason, on the fifth day of hospitalization, treatment with remdesivir was started.

After the administration of CP and then RDV, there was a progressive reduction in the viral load on BAL (see Table 2, 1.31 × 10^9^ at hospitalization day 5 to 1.75 × 10^6^ copies/mL at hospitalization day 13).

Today, we would use monoclonal antibodies instead of CP, specifically etesevimab plus bamlanivimab that on 3 December 2021 received emergency use authorization from the FDA for the pediatric age, including newborns [20].

In order to investigate the state of alteration of consciousness and of the spontaneous motility, a cerebral TC scan with mdc was performed and was normal. Considering the persistent impairment of the neurological condition, we completed the radiological evaluation with a brain MRI that revealed the presence of thrombosis of the deep medullary veins and, for this reason, heparin was started at a therapeutic dosage. According to Magro and Beslow, low molecular weight heparin is a possible treatment in cerebral thrombosis during COVID-19 [21,22].

Considerable evidence suggests that COVID-19 is associated with a hypercoagulable state. This is reflected in the extremely elevated D-dimer levels (a marker of clot turnover) observed in many patients, including children, over the course of the first few weeks of disease, particularly those who are more severely affected [22,23]. Moreover the significant increase in NT-proBNP shown by our patient seems to confirm the essential role of the NO/cGMP axis, ultimately leading to thrombotic events [24,25].

Fletcher-Sandersjöö et al. attributes the hypercoagulability to an overactivation of the complement system, although the pathophysiology of coagulopathy in SARS-CoV-2 infection is still unknown [26].

After thirteen days of hospitalization, the baby was extubated and in four more days became oxygen independent, able to feed on the breast, with neurological and EEG improvement. A control brain MRI was repeated at five months from the previous one and showed a complete resolution of the thrombotic lesions.

## 4. Conclusions

The present case report describes the possible treatment for severe SARS-CoV-2 infection in newborns. To date, the therapy for COVID-19 infection in newborns is not supported by specific guidelines, and, therefore, the treatment is defined case by case on the basis of the few case reports available [8,9]. CTV, a rare and severe complication related to SARS-CoV-2 infection, represents a further challenge for the clinician especially in the neonatal period. Further investigations are needed to guide treatment in infants with COVID-19.

## Figures and Tables

**Figure 1 ijerph-18-13201-f001:**
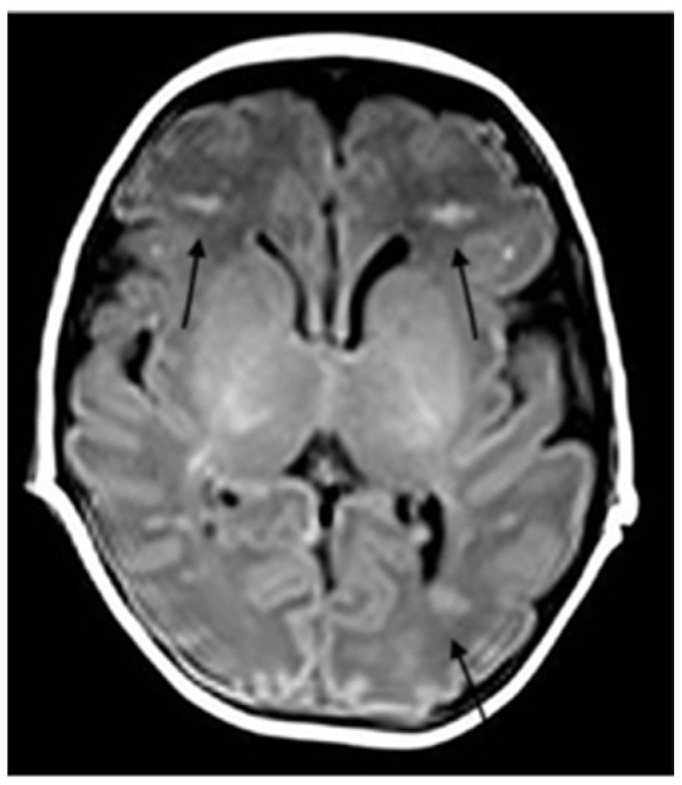
Axial T1-weighted section shows bilateral hyperintense periventricular WM linear lesions suggestive of thrombotic DMVs (arrows).

**Figure 2 ijerph-18-13201-f002:**
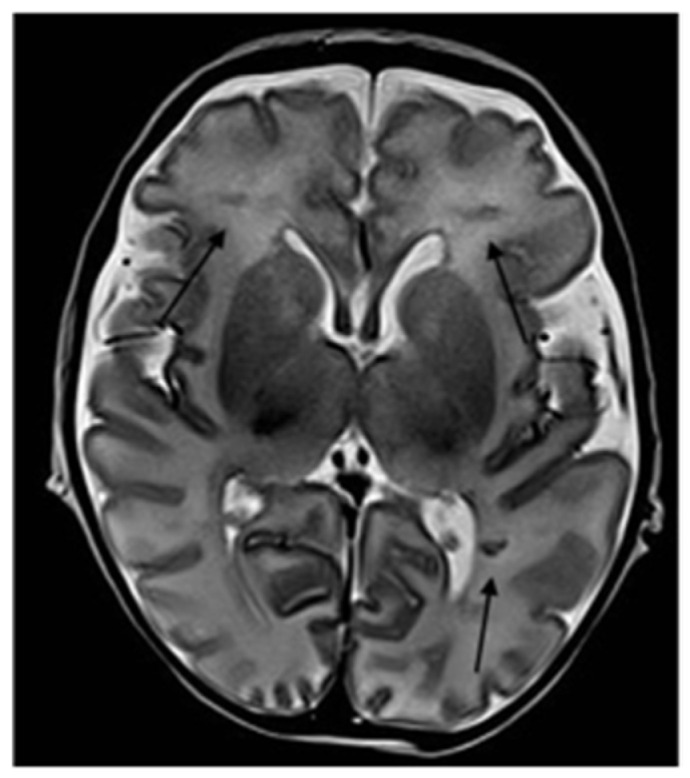
Axial T2-weighted section shows bilateral periventricular hypointense WM linear lesions (arrows).

**Figure 3 ijerph-18-13201-f003:**
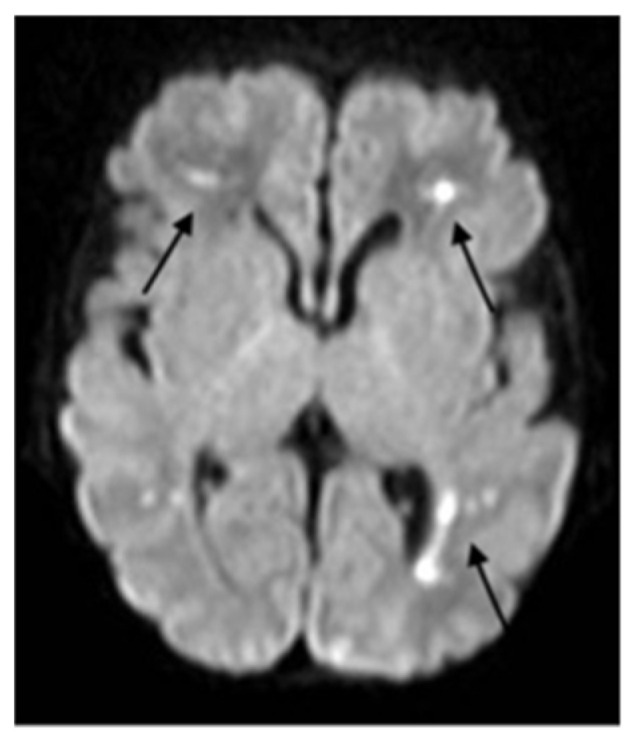
Axial DWI section shows significantly increased of signal (cytotoxic edema) in frontal and posterior pertrigonal areas (arrows).

**Figure 4 ijerph-18-13201-f004:**
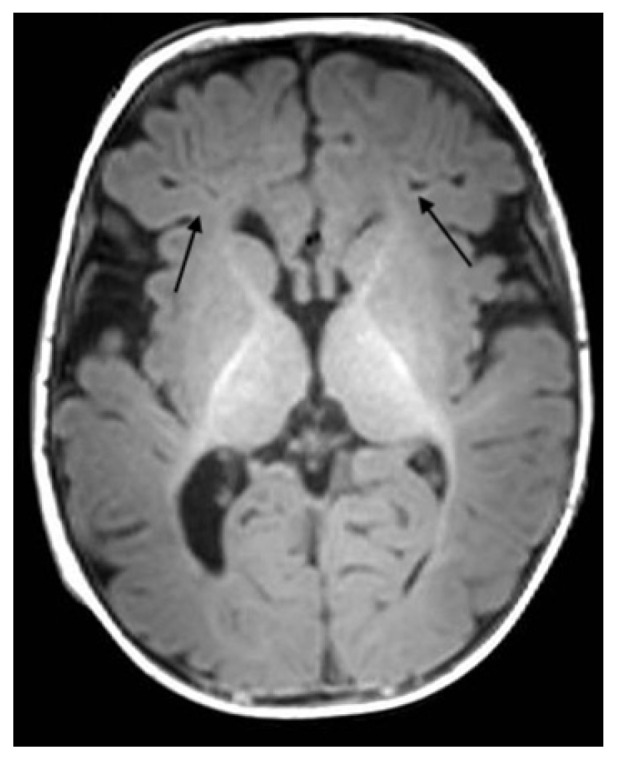
Axial T1-weighted section in follow-up MRI examination at 5 months demonstrate reduction of WM linear hypointesity in frontal WM (arrows).

**Figure 5 ijerph-18-13201-f005:**
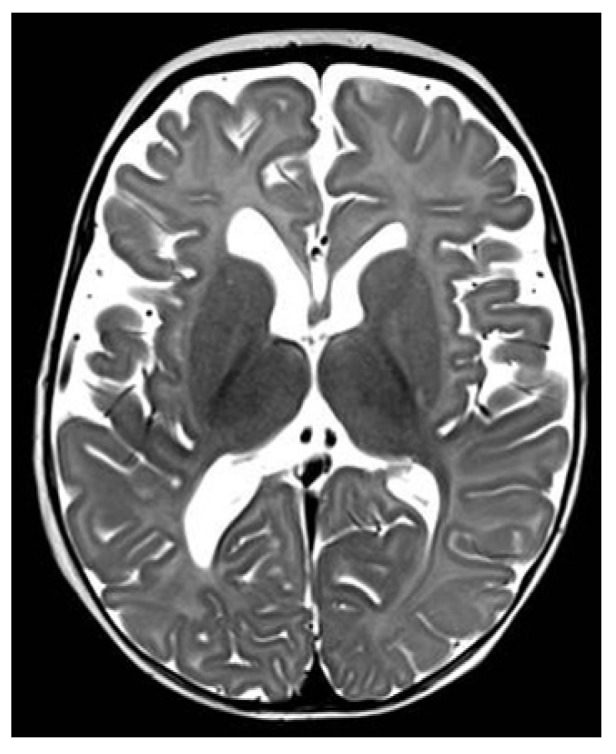
Axial T2-weighted section in follow-up MRI examination at 5 months: normal signal of periventricular WM (note the enlargement of lateral ventricles).

**Figure 6 ijerph-18-13201-f006:**
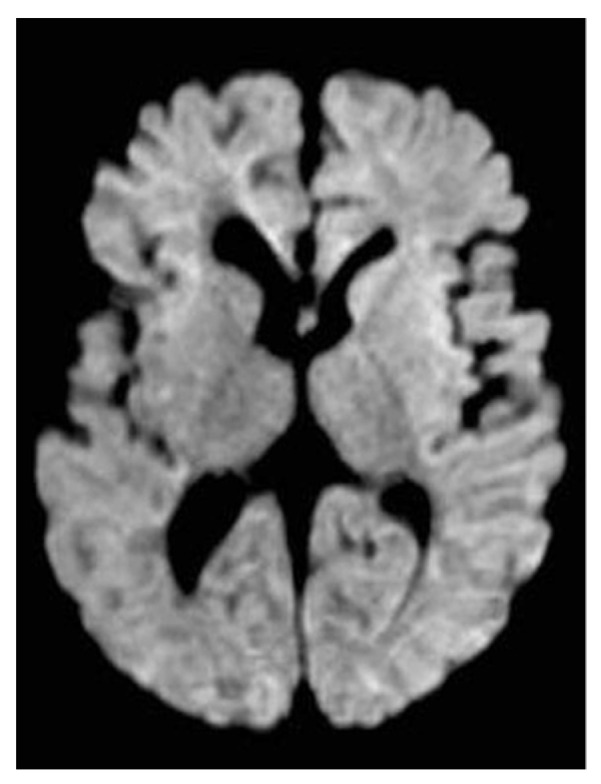
Normal signal in DWI in follow-up MRI examination at 5 months.

**Table 1 ijerph-18-13201-t001:** Laboratory findings.

Laboratory Findings	Normal Range	Day 1 after Admission	Day 5 after Admission	Day 10 after Admission
WBC count per mm^3^	5000–19,500	3230	1830	8960
Lymphocyte count per mm^3^	2000–14,620	610	480	1450
Hemoglobin in g/dL	12.5–20.5	16.8	8.6	9.8
Platelet per mm^3^	150,000–450,000	181,000	172,000	441,000
C-Reactive protein mg/dL	<0.5	0.11	0.05	0.1
Procalcitonin ng/mL	<0.5	1.14	0.6	0.31
Troponina T high sensitive ng/L	<14	54.8	77.6	41.6
N-Terminal prohormone of Brain Natriuretic Peptide pg/mL	<320	3702	8115	5145
D-Dimer mg/L	<0.5	1.77	0.76	2.41
Fibrinogen mg/dLA	283–401	197	67	146
Ferritin ug/mL	30–400	1000	659	402
Lactic Dehydrogenase U/L	225–600	520	234	317
Serum albumin g/dL	3.8–5.4	2.8	3.9	4.8

**Table 2 ijerph-18-13201-t002:** SARS-CoV-2 viral load determination.

Specimen	Sampling (dd/mm/aa)	SARS-CoV-2 Viral Load (cp/mL log)
NPS	06/03/21	1.17 × 10^10^
BAL	10/03/21	2.68 × 10^9^
BAL	13/03/21	1.31 × 10^9^
BAL	16/03/21	1.29 × 10^6^
BAL	19/03/21	1.42 × 10^5^
BAL	21/03/21	1.75 × 10^6^
NPS	26/03/21	1.44 × 10^3^
NPS	10/04/21	1.12 × 10^3^

NPS: nasopharyngeal swab; BAL: bronchoalveolar lavage.

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
