# Peer review of "Severe COVID-19 Complicated by Cerebral Venous Thrombosis in a Newborn Successfully Treated with Remdesivir, Glucocorticoids, and Hyperimmune Plasma"

_ijerph, 2021, doi:10.3390/ijerph182413201_

Round 1

Reviewer 1 Report

I would like to congratulate the team on its prompt emergency medical treatment of this newborn boy, and to a very good written case report! When it comes to laboratory findings, the large increase in NT-proBNP seems to confirm the essential role of the NO/cGMP axis.  

Author Response

Thank you very much for your time and your suggestions! We carefully review our english and we add a sentence (and 2 references) in discussion to comment on the elevation of pro-BNP ant the role of NO/cGMP axis in the thrombotic events (pag 4 lines 183-185, reference 22-23)

Reviewer 2 Report

The authors are to congratulate for the presentation of this case report. Serious SARS-CoV-2 infection in children is rare although it has been shown that a U-shaped curve of severity exists in children diagnosed with COVID-19, and infants are at a higher risk of developing severe disease. Thus, especially complicated cases like this one are valuable. 

I have one major comment:

The authors should explicitly name the syndrome as Multisystem Inflammatory Syndrome in Children since there are features of fever, SARS-CoV-2 infection, and at least 2 system involvement (gastrointestinal and neurological). Recently, the American College of Rheumatology published relevant guidelines (https://pubmed.ncbi.nlm.nih.gov/33277976/) and authors should discuss them. Since the pandemic is ongoing and there are constant updates on the clinical treatment, the authors should extensively discuss what the clinical treatment would be now, for example, use of monoclonal antibodies or COVID-specific antivirals such as ritonavir or molnupiravir.

Minor text editing comments

Line 68: "a high" instead of "an high"

Line 84: "breast" instead of "brest"

Line 95: remove one dot from the end

Author Response

Dear reviewer, we would like to thank you for your time and for the interesting comments that definitely improve our paper. We introduce in the discussion  the differential diagnosis with MIS-C and we argumented why we did use a treatment that differs from what recommended by the rheumatological  guidelines for MIS-C (pag 3, lines 127-135) . Moreover, as suggested, we discuss what we woud have done differently nowadays (pag 4, lines 170-172).

Last but not least, we carefully review the paper for spelling/typing errors. 

Once again, thank you for your revision

Round 2

Reviewer 2 Report

No further comments